# The Diagnostic Accuracy of Procalcitonin and Its Combination with Other Biomarkers for Candidemia in Critically Ill Patients

**DOI:** 10.3390/jcm13123557

**Published:** 2024-06-18

**Authors:** Stelios Kokkoris, Epameinondas Angelopoulos, Aikaterini Gkoufa, Foteini Christodouli, Theodora Ntaidou, Evangelia Theodorou, Georgia Dimopoulou, Ioannis Vasileiadis, Panagiotis Kremmydas, Christina Routsi

**Affiliations:** First Department of Critical Care Medicine and Pulmonary Services, Medical School, National and Kapodistrian University of Athens, Evangelismos Hospital, 45-47 Ipsilantou Street, 10676 Athens, Greece; skokkoris2003@yahoo.gr (S.K.); angelopoulosdre@yahoo.it (E.A.); katergouf@yahoo.gr (A.G.); foteini.christodouli@nhs.net (F.C.); theodorantaidou@yahoo.com (T.N.); evaggeliatheodorou@gmail.com (E.T.); ginadim@outlook.com (G.D.); ioannisvmed@yahoo.gr (I.V.); kremmydasp19@gmail.com (P.K.)

**Keywords:** procalcitonin, C-reactive protein, NLR, sepsis, bacteremia, candidemia, ICU, biomarkers

## Abstract

**Background:** The aim of this study was to investigate the usefulness of serum procalcitonin (PCT), C-reactive protein (CRP), neutrophil to lymphocyte count ratio (NLR), and their combination, in distinguishing candidemia from bacteremia in intensive care unit (ICU) patients. **Methods:** This is a retrospective study in ICU patients with documented bloodstream infections (BSIs) and with both serum PCT and CRP measurements on the day of the positive blood sample. Illness severity was assessed by sequential organ failure assessment (SOFA) score on both admission and BSI day. Demographic, clinical, and laboratory data, including PCT and CRP levels and NLR on the day of the BSI, were recorded. **Results:** A total of 63 patients were included in the analysis, of whom 32 had bacteremia and 31 had candidemia. PCT, CRP, and NLR values were all significantly lower in candidemia compared with bacteremia (0.29 (0.14–0.69) vs. 1.73 (0.5–6.9) ng/mL, *p* < 0.001, 6.3 (2.4–11.8) vs. 19 (10.7–24.8) mg/dl, *p* < 0.001 and 6 (3.7–8.6) vs. 9.8 (5.3–16.3), *p* = 0.001, respectively). PCT was an independent risk factor for candidemia diagnosis (OR 0.153, 95%CI: 0.04–0.58, *p* = 0.006). A multivariable model consisting of the above three variables had better predictive ability (AUC-ROC = 0.88, *p* < 0.001), for candidemia diagnosis, as compared to that of PCT, CRP, and NLR, whose AUC-ROCs were all lower (0.81, *p* < 0.001, 0.78, *p* < 0.001, and 0.68, *p* = 0.015, respectively). **Conclusions:** A combination of routinely available laboratory tests, such as PCT, CRP, and NLR, could prove useful for the early identification of ICU patients with candidemia.

## 1. Introduction

Intensive care unit (ICU)-acquired bloodstream infections (BSIs) frequently complicate the course of critically ill patients and they are associated with increased morbidity and mortality [1,2]. In addition to bacterial pathogens, fungi, predominantly the *Candida* species, cause a notable proportion of BSIs with their prevalence increasing worldwide [3,4,5,6].

Early and empirical initiation of appropriate treatment in patients with suspected BSIs is strongly recommended [7], due to its association with a better prognosis [7,8,9,10]. However, early distinction (i.e., prior to the identification of blood pathogens by the microbiology laboratory) of the bacterial or fungal type of a BSI is not feasible, since both types of infection share similar clinical features. Nonetheless, early suspicion, guided by laboratory parameters, could be of great importance in distinguishing the types of BSIs, and thereby offering more individualized treatment. Several laboratory biomarkers indicative of infection, including white blood cells (WBCs) [11] and WBC differential counts [12,13], C-reactive protein (CRP) [14], and procalcitonin (PCT) [15,16], are routinely available in clinical practice and could serve as adjunctive tools in this context.

Particularly, PCT, a prohormone that is the precursor of calcitonin, is a widely used marker that accurately differentiates systemic bacterial infections from noninfectious inflammatory states in ICU patients [14,15,16]. However, in the context of BSIs, the PCT response to *Candida* species has been less investigated than that to bacterial pathogens. Several studies have suggested a potential role of PCT in discriminating between bacteremia and candidemia in both hospital wards [17,18,19] and the ICU [20,21,22,23,24,25]. High PCT values were predominately associated with bacteremia mainly caused by Gram-negative pathogens, while low PCT values successfully ruled out BSI as a cause of sepsis [26]. Similarly, the highest PCT levels were observed in Gram-negative bacteremia with only marginal increases in fungal infection elsewhere [27]. However, contrasting data have been reported, indicating increased levels of PCT in ICU patients with candidemia [28].

Although a recent systematic review [29] confirmed that serum PCT concentrations were higher in patients with bacteremia compared to those with candidemia, it was recommended that PCT should not be used as a standalone tool for the differential diagnosis between candidemia and bacteremia, due to insufficient supporting evidence. Most of the aforementioned studies have focused exclusively on PCT as a sole biomarker.

Apart from PCT, the information concerning other individual markers of infection as discrimination tools between candidemia and bacteremia is, thus far, scarce. Therefore, there is limited evidence regarding the effectiveness of PCT in combination with biomarkers, such as CRP [22,23,25], WBC count [23,26], or WBC differential, in distinguishing between bacterial and fungal type of BSIs.

The objective of this study was to investigate the usefulness of serum PCT, CRP, neutrophil to lymphocyte count ratio (NLR), as well as their combination, in distinguishing bacterial versus *Candida* species etiology of BSIs in ICU patients.

## 2. Methods

### 2.1. Data Collection

This is a retrospective clinical study, conducted in the ICU of the Evangelismos Hospital, a 1000-bed teaching hospital in Athens, Greece. We searched the hospital electronic record system from April 2012 to October 2014 to identify patients admitted to the ICU who developed BSIs and, additionally, had concomitant measurements of PCT, CRP, and WBC count on the day of blood culture collection. In our ICU, CRP and WBC count measurements were performed as part of daily routine care, whereas PCT assessment was performed only if it was ordered by the attending physician in patients with clinically suspected sepsis. Patients with neutropenia, hematological malignancy, or other hematological disease were excluded. Demographics, date of ICU admission, date of BSI, detected pathogen, admission diagnosis classified as medical or surgical, illness severity, length of stay in ICU, and ICU clinical outcome were recorded. The severity of acute illness was evaluated by the Acute Physiology and Chronic Health Evaluation (APACHE) II score [30] on ICU admission. The severity of organ dysfunction was assessed by the Sequential Organ Failure Assessment (SOFA) score [31], calculated on the first day of ICU admission and, additionally, on the day of BSI. The characteristics of ICU patients who developed bacteremia were compared with those who developed candidemia during their ICU stay. The approval for the use of the de-identified data was obtained from the Ethics Committee of the hospital (protocol number 241/2024). An informed consent from the patient’s next-of-kin was waived.

### 2.2. Laboratory Methods

CRP was measured by an immunoturbidimetric assay for the in vitro quantitative determination of CRP in human serum on Roche/Hitachi cobas c systems, Roche Diagnostics (Hellas) S.A., Athens, Greece (measuring range 0.30–40.0 mg/dl, normal values < 0.5 mg/dl). PCT was measured by the Elecsys^®^ BRAHMS PCT assay (ElectroChemiLuminescence), Roche Diagnostics (Hellas) S.A., Atens, Greece (limit of detection 0.02 ng/mL, measuring ranges 0.02–100 ng/mL). NLR was calculated by dividing the absolute neutrophil count with the absolute lymphocyte count.

### 2.3. Microbiological Methods

The BD Bactec (Becton Dickinson, Sparks, MD, USA) automated blood culture system was used for monitoring blood culture bottles. Bacterial and fungal isolates were identified at species level using the VitekMS (BioMeriéux, Marcy l’Étoile, France) device and the MALDI-TOF MS method.

### 2.4. Definitions

ICU-acquired BSI was defined as that occurring in an ICU patient when one or more cultures of blood, obtained more than 48 h after admission to the ICU, yielded a pathogenic microorganism. Blood culture specimens were ordered by the attending physicians in the presence of clinical features compatible with sepsis or septic shock [11], or when infection was suspected on clinical rounds. The onset of BSI was defined as the date of the blood sampling.

Bacteremia was defined as the recovery of any bacterial species, whereas candidemia was defined as the presence of any *Candida* species in the blood specimen. When both *Candida* and bacteria strains were simultaneously grown by the same blood culture, the BSI was not included in the analysis.

### 2.5. Statistical Analysis

Quantitative data are reported as median and interquartile range (IQR). Qualitative data are reported as number (%). Non-parametric statistical tests were applied, due to the non-normal distribution of the data, as determined by the Kolmogorov–Smirnov test. Comparisons between patients who developed bacteremia vs. candidemia in the ICU were performed using the Mann–Whitney U test. Differences between these two groups of patients in qualitative variables were assessed by Chi-square or Fisher’s exact test when appropriate. Logistic regression analysis was used to build a multivariable model and adjust for confounders. All variables (PCT, CRP, NLR) were log-transformed before entering the model. Receiver operating characteristics–areas under the curve (ROC-AUCs) were used to estimate the predictive performance of various parameters for candidemia diagnosis. Predicted probabilities of the multivariable model, as estimated by the logistic regression, were used to calculate the ROCs-AUCs of the model. Cut-off values for NLR, PCT, CRP, and the model’s predicted probabilities were estimated by Youden’s index. Diagnostic characteristics, such as sensitivity, specificity, positive and negative predictive value (PPV and NPV), and positive and negative likelihood ratio (LR+ and LR−), for the above cut-off values were evaluated. The SPSS statistical program (version 24, Chicago, IL, USA) was used for data analysis. Statistical significance was defined as a two-tailed *p* value of <0.05.

## 3. Results

### 3.1. Patient Characteristics

A total of 63 patients were included in the analysis, of whom 32 (51%) had bacteremia and 31 (49%) had candidemia. The median (IQR) age was 60 (41–74) years, 66% were males, and the crude ICU mortality was 35%. Admission APACHE II and SOFA scores were 20 (15–23) and 10 (5–11), respectively (Table 1). *Klebsiella pneumoniae* (18%) and *Acinetobacter baumannii* (8%) were the most frequent bacteria, whereas *Candida parapsilosis* (21%) and *Candida albicans* (13%) were the most frequent *Candida species* (Table 2). NLR, PCT, and CRP were all significantly lower in candidemia compared with bacteremia (6 (3.7–8.6) vs. 9.8 (5.3–16.3), *p* = 0.001, 0.29 (0.14–0.69) vs. 1.73 (0.5–6.9) ng/mL, *p* < 0.001, and 6.3 (2.4–11.8) vs. 19 (10.7–24.8) mg/dl, *p* < 0.001, respectively), see Table 1 and Figure 1.

### 3.2. Multivariable Logistic Regression Model

We performed a multivariable logistic regression analysis, which revealed PCT as an independent risk factor for candidemia diagnosis (OR: 0.153, 95%CI: 0.04–0.58, *p* = 0.006), Table 3. Specifically, the higher its levels, the lower the odds of candidemia.

### 3.3. Diagnostic Characteristics of PCT, CRP, and NLR for Candida BSI

ROC curves were constructed for PCT, CRP, NLR, and the multivariable model, Figure 2 Cut-off values for NLR, PCT, CRP, and the model’s predicted probabilities were estimated by Youden’s index. Then, we calculated all diagnostic characteristics for candidemia prediction for each variable. PCT and CRP had low sensitivity, NPV and LR− (PCT: 0.58, 0.69 and 0.46, CRP: 0.71, 0.75 and 0.34, respectively), but high specificity, PPV and LR+ (PCT: 0.91, 0.86, and 6.18, CRP: 0.84, 0.81, and 4.55, respectively) for candidemia diagnosis. The multivariable model mentioned above (built by the combination of NLR, PCT, and CRP), had a better overall diagnostic performance, showing higher sensitivity and NPV (0.93 and 0.92, respectively), as well as high specificity, PPV, and LR+ (0.77, 0.79, and 4.11, respectively). Furthermore, it had better predictive ability (AUC-ROCs = 0.88, *p* < 0.001) for candidemia diagnosis, as compared to that of PCT, CRP, and NLR, whose AUC-ROCs were all lower (0.81, *p* < 0.001, 0.78, *p* < 0.001, and 0.68, *p* = 0.015, respectively), Table 4, Figure 2.

## 4. Discussion

We conducted a retrospective study in septic ICU patients with BSIs seeking to determine routine laboratory tests which could early differentiate candidemia from bacteremia. The main findings of the present study could be summarized as follows: firstly, PCT, CRP, and NLR values were all significantly lower in candidemia compared with bacteremia. Secondly, PCT was an independent risk factor for candidemia diagnosis, after adjustment for the two other biomarkers, CRP and NLR. Third, PCT and CRP had high specificity, PPV, and LR+, but low sensitivity, NPV, and LR−, for candidemia diagnosis. Lastly, the combination of all three parameters had even better diagnostic characteristics, as well as overall predictive performance for candidemia diagnosis (as estimated by ROC-AUCs), compared to each parameter individually.

A recent systematic review [29] summarized the current evidence about PCT values for differentiating candidemia from bacteremia including 16 studies with a total of 45,079 patients and 785 cases of candidemia. Of these studies, 10 exclusively referred to ICU patients. However, the studies identified were clinically very heterogeneous and involved different assessment methods. Although the majority of these studies showed lower PCT values in patients with candidemia compared to bacteremia, the evidence supporting this observation was of low quality and the differences seemed insufficiently discriminative to guide therapeutic decisions. What the present study adds to the field is that the combination of PCT, CRP, and NLR had superior diagnostic characteristics, as well as overall predictive performance, compared to each one of them separately, for the early diagnosis of candidemia, Table 4.

Only a few studies addressed the role of PCT in combination with other biomarkers in discriminating between bacterial and *Candida* BSIs [22,24,25,26,32]. In accordance with our findings, Fu et al. [22], found that the combination of PCT (cut-off 8.06 ng/mL), CRP (cut-off value 11.6 mg/dl), and IL-6 (cut-off 186.5 pg/mL) increased the sensitivity and specificity for early diagnosis of candidemia and its distinction from Gram-positive/negative bacteremia (AUC = 0.912). In contrast, Martini et al. [24], in a cohort of surgical ICU patients, found that the combination of CRP (with a cut-off value of 10 mg/dl) and PCT (with a cut-off of 2 ng/mL) did not increase sensitivity or specificity for a diagnosis of *Candida* sepsis. Another study [32] reported improved diagnostic performance for the combination of PCT with (1,3)-β-D-glucan for candidemia.

Due to our local ICU ecology, as previously reported [33], Gram-negative pathogens predominated in the microbiology pattern of BSIs (Table 2). Therefore, in the absence of BSIs caused by Gram-positive pathogens, comparisons were made between candidemia and Gram-negative bacteremia. It should be noted that low levels of PCT in the presence of Gram-positive BSIs have been demonstrated elsewhere [21].

Both bacterial and fungal infections usually cause an increase in the counts of neutrophils, and hence an increase in NLR. NLR is considered a reliable marker for the diagnosis of bacteremia and sepsis and has a potential value in assessing the severity of sepsis [34]. However, to our knowledge, there is a paucity of studies which assess the usefulness of NLR in distinguishing the bacterial or the *Candida* species etiology of BSIs in ICU patients. We found that NLR was significantly lower in candidemia compared to bacteremia patients. A cut-off value of 12.67 had high sensitivity (0.92), NPV (0.86) and LR+ (1.60), but low specificity (0.41), PPV (0.59) and LR− (0.17). In line with our results, Marik et al. [26] found that the candidemia group had lower NLR compared to that of other pathogens or control groups.

One of the main findings of this study is that in ICU patients with sepsis or septic shock, serum PCT is not markedly elevated at the onset of candidemia, whereas high values are mainly found in bacteremia cases. Similar findings have been reported by several studies so far [20,21,22,23,24,25,27,29,32]. Interestingly, we also found that this occurred independently of the level of systemic inflammation as assessed by CRP levels, as well as of other possible confounding variables, such as NLR. Additionally, a low cut-off value of 0.34 ng/mL had low sensitivity, NPV, and LR− but high specificity, PPV, and LR+ for candidemia diagnosis (Table 4).

To interpret the difference in PCT values between candidemia and bacteremia cases, we must take into account several factors. First, the PCT expression is, at least in part, induced by pro-inflammatory cytokines [35]. The differences in the pathogen-specific signaling could explain the observed differences in the present as well as in other relevant studies. In this context, fungal infection may trigger a different pattern and/or magnitude of cytokine response compared to that encountered during bacterial infections [23,35]. Secondly, a time-related different impairment of the host immune response in the patients with candidemia compared with those with bacteremia may be responsible [23]. Indeed, candidemia occurred significantly later during the ICU stay than did bacteremia in our population (24 vs. 14 days from ICU admission to BSI onset, respectively), which is consistent with the assumption of immune paralysis-related candidemia, due to antecedent septic episodes or any other condition that could inflict upon systemic inflammation.

CRP is a well-known sepsis biomarker [14]. In the present study, it was found to have similar performance to PCT with respect to candidemia diagnosis. Specifically, it was significantly lower in candidemia compared with bacteremia. However, it was not found to be an independent risk factor for candidemia diagnosis. A low cut-off value of 9.75 mg/dl had moderate sensitivity, NPV, and LR−, but a high specificity, PPV, and LR+ for candidemia diagnosis (Table 4). There is limited and conflicting evidence regarding the diagnostic accuracy of CRP in the setting of candidemia in the ICU [22,36,37]. Since PCT may be considered as an acute phase protein like CRP [38], the explanations provided above for the PCT could also apply to CRP.

Certain limitations of the present study should be pointed out. First, since not all consecutive BSIs have been included in the analysis but only those with concomitant PCT measurement, there could be a possibility for selection bias. Second, mixed etiology of BSIs, i.e., bacteremia with concomitant candidemia, were not included. Third, PCT values in cases of negative blood cultures are not available, since only microbiologically documented BSIs have been included. For this reason, a low PCT value should be used cautiously as a single biomarker for early diagnosis of candidemia, since, as expected, negative blood cultures are also associated with low PCT levels [27]. In addition, due to the retrospective type of this single-center study and because of the small number of the included patients, our results cannot be extrapolated. However, this study clearly demonstrates that the combination of easily accessible, everyday clinical laboratory markers had a good diagnostic performance for *Candida* species BSIs, which could be important for resource-limited regions.

Finally, considering the results of the present study along with those previously reported, it should be noted that, despite the complexity and the limitations of PCT, ICU clinicians, apart from the significance of increased PCT values, should also take into account the low PCT values in the presence of sepsis or septic shock.

In conclusion, PCT was found to be independently associated with candidemia diagnosis in septic ICU patients with BSIs. Specifically, the higher its levels, the lower the odds of candidemia. In addition, its combination with the routinely available biomarkers, CRP and NLR, was superior to early identification of those patients with candidemia. Further studies are required to delineate whether these results might impact current therapeutic algorithms, concerning those patients who would benefit from early appropriate treatment.

## Figures and Tables

**Figure 1 jcm-13-03557-f001:**
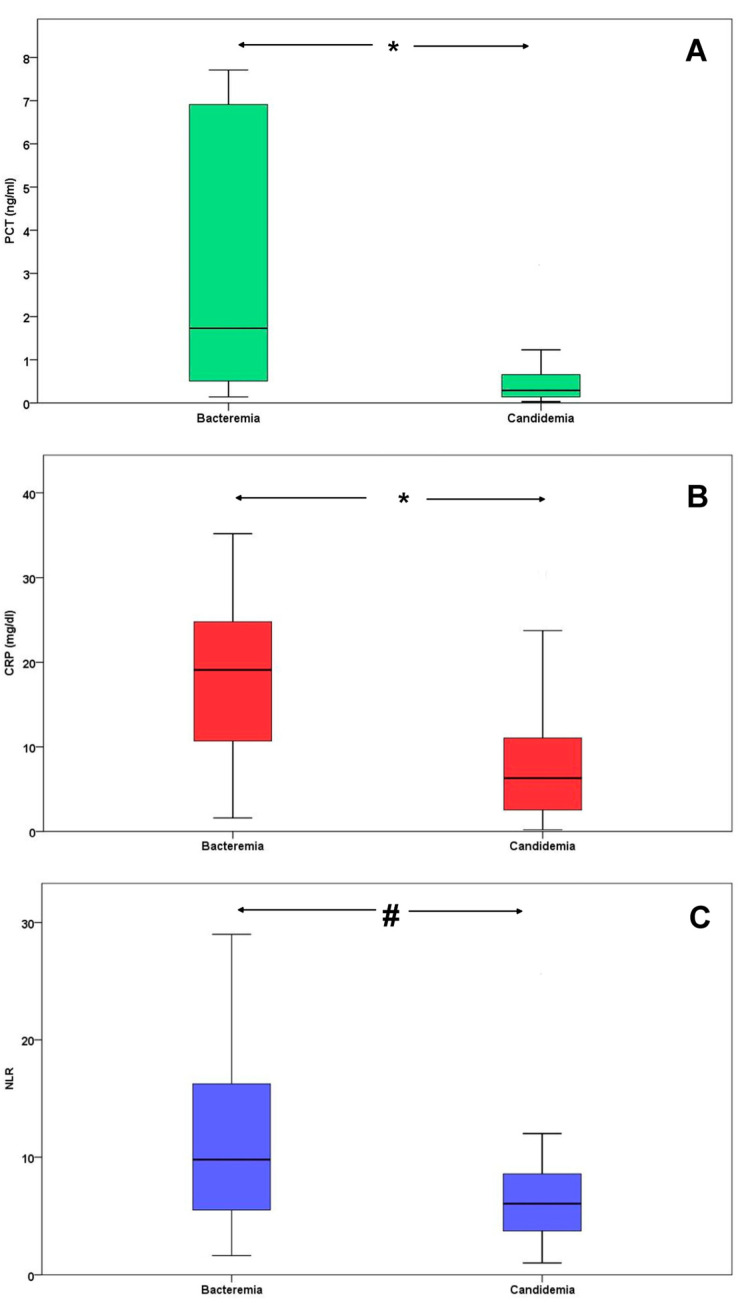
Boxplots of (**A**) PCT, (**B**) CRP, and (**C**) NLR, in patients with bacteremia and candidemia.* *p* < 0.001, # *p* = 0.015. Abbreviations: PCT, procalcitonin; CRP, C-reactive protein; NLR, neutrophil to lymphocyte ratio.

**Figure 2 jcm-13-03557-f002:**
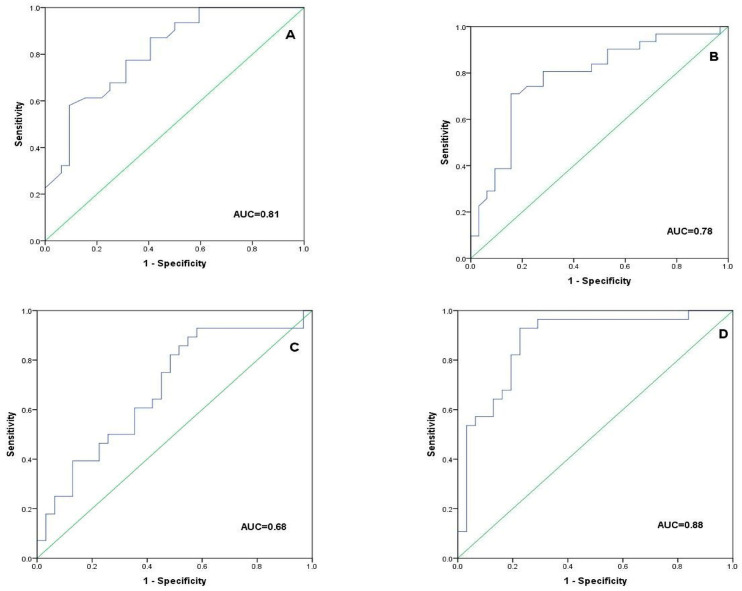
ROC curves for candidemia diagnosis of (**A**) PCT, (**B**) CRP, (**C**) NLR, and (**D**) a multivariable model consisted of the combination of all three markers. Abbreviations: ROC, receiver operating characteristic; PCT, procalcitonin; CRP, C-reactive protein; NLR, neutrophil to lymphocyte ratio.

**Table 1 jcm-13-03557-t001:** Patient baseline characteristics and laboratory tests on day of BSI occurrence.

	All Patients (*n* = 63)		Bacteremia (*n* = 32)		Candidemia (*n* = 31)		*p*- Value
Age, years	60	(41–74)	60	(44–66)	63	(39–75)	0.62
Sex (male), *n* (%)	41	(66)	21	(68)	20	(64)	0.78
Admission category							0.26
Surgical, *n* (%)	28	(44)	12	(37)	16	(52)	
Medical, *n* (%)	35	(56)	20	(62)	15	(48)	
Outcomes							
ICU mortality, *n* (%)	22	(35)	14	(44)	8	(26)	0.32
ICU-LOS, days	42	(26–58)	43	(24–57)	35	(26–82)	0.77
Severity scores							
SOFA on admission	10	(5–11)	9	(5–11)	10	(6–12)	0.66
APACHE II on admission	20	(15–23)	20	(15–24)	19	(15–22)	0.59
SOFA on BSI day	8	(5–11)	10	(6–12)	6	(4–10)	0.08
BSI day since admission, days	24	(10–42)	14	(5–29)	24	(15–42)	0.16
Laboratory tests on BSI day							
WBCs, ×10^9^/l	11.79	(8.47–18.38)	17.09	(10.86–22.83)	8.93	(6.77–12.82)	**<0.001**
Lymphocytes, ×10^9^/l	1.41	(0.94–1.84)	1.47	(1.01–1.94)	1.38	(0.88–1.77)	0.42
PMN, ×10^9^/l	8.56	(5.96–15.57)	13.08	98.10–19.13)	6.30	(5.71–10.05)	**0.001**
NLR	7.06	(4.56–13.34)	9.79	(5.29–16.26)	6.04	(3.71–8.58)	**0.015**
CRP, mg/dl	10.7	(4.3–20.9)	19.0	(10.7–24.7)	6.3	(2.4–11.8)	**<0.001**
PCT, ng/mL	0.62	(0.24–2.85)	1.73	(0.51–6.91)	0.29	(0.14–0.69)	**<0.001**

Data are expressed as median (IQR), unless otherwise defined. Abbreviations: BSI, blood stream infection; ICU, intensive care unit; LOS, length of stay; SOFA, sequential organ failure assessment; APACHE, acute physiology and chronic health evaluation; WBCs, white blood cells; PMN, polymorphonuclears; NLR, neutrophil to lymphocyte ratio; CRP, C-reactive protein; PCT, procalcitonin; IQR, interquartile range.

**Table 2 jcm-13-03557-t002:** Candidemia and bacteremia species.

	*n*	%
Candidemia species		
*Candida parapsilosis*	13	20.6
*Candida albicans*	8	12.7
*Candida glabrata*	3	4.8
*Candida tropicalis*	1	1.6
*Candida luzitanea*	1	1.6
*Candida crusei*	1	1.6
Bacteremia species		
*Klebsiella pneumoniae*	11	17.5
*Acinetobacter baumanni*	5	7.9
*Pseudomonas aeruginosa*	5	7.9
*Providencia stuartii*	5	7.9
Other bacteria/fungi	9	14

**Table 3 jcm-13-03557-t003:** Multivariable logistic regression model for candidemia diagnosis.

Variable	B Coefficient	*p*-Value	OR	95%CI
Log[PCT]	−1.879	0.006	0.153	0.040–0.580
Log[CRP]	−1.190	0.205	0.304	0.048–1.914
LogNLR	−1.881	0.097	0.152	0.016–1.408

Brackets denote concentration. Abbreviations: OR, odds ratio; CI, confidence interval; PCT, procalcitonin; CRP, C-reactive protein; NLR, neutrophil to lymphocyte ratio.

**Table 4 jcm-13-03557-t004:** Diagnostic tests of PCT, CRP, NLR, and their combination, for candidemia.

Cut-Off Value *	PCT = 0.34 ng/mL	CRP = 9.75 mg/dl	NLR = 12.67	Model’s Predicted Probability = 0.40
Sensitivity	0.58 [0.41–0.73]	0.71 [0.53–0.84]	0.92 [0.77–0.98]	0.93 [0.77–0.98]
Specificity	0.91 [0.76–0.97]	0.84 [0.68–0.93]	0.41 [0.26–0.59]	0.77 [0.60–0.88]
PPV	0.86 [0.65–0.95]	0.81 [0.63–0.92]	0.59 [0.44–0.72]	0.79 [0.62–0.89]
NPV	0.69 [0.54–0.81]	0.75 [0.59–0.86]	0.86 [0.62–0.96]	0.92 [0.76–0.98]
LR+	6.18 [2.02–18.94]	4.55 [1.97–10.48]	1.60 [1.16–2.19]	4.11 [2.12–7.95]
LR−	0.46 [0.30–0.71]	0.34 [0.19–0.61]	0.17 [0.04–0.69]	0.09 [0.02–0.35]
ROC-AUC	0.81 [0.71–0.92] ^#^	0.78 [0.67–0.90] ^#^	0.68 [0.55–0.82] ^$^	0.88 [0.78–0.97 ] ^#^

Data within brackets indicate 95% confidence interval. *, Cut-off values for NLR, PCT, CRP, and the model’s predicted probabilities were estimated by Youden’s index. The model consisted of the combination of PCT, CRP, and NLR. ^#^, *p* < 0.001, ^$^, *p* = 0.015. Abbreviations: PCT, procalcitonin; CRP, C-reactive protein; NLR, neutrophil to lymphocyte ratio; PPV, positive predictive value; NPV, negative predictive value; LR+, positive likelihood ratio; LR−, negative likelihood ratio; ROC-AUC, receiver operating characteristics–area under the curve.

## Data Availability

The datasets used/or analyzed in the present study are available from the corresponding author upon reasonable request.

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
