# Peer review of "The Diagnostic Accuracy of Procalcitonin and Its Combination with Other Biomarkers for Candidemia in Critically Ill Patients"

_jcm, 2024, doi:10.3390/jcm13123557_

Round 1

Reviewer 1 Report

Comments and Suggestions for Authors

The manuscript is well written and contains relevant information for the management of critically ill patients. However, some points need to be better clarified to facilitate reader understanding.

Table 2 - Scientific names

The species Candida glabrata and Candida krusei are currently called Nakazeomyces glabrata and Pichia kudriavzevii, respectively.

What techniques were used to identify the species. It is necessary to describe the methods.

In the discussion, I think it is important to further explore the expression of cytokines related to septic shock.

Author Response

The manuscript is well written and contains relevant information for the management of critically ill patients. However, some points need to be better clarified to facilitate reader understanding.

Table 2 - Scientific names

The species Candida glabrata and Candida krusei are currently called Nakazeomyces glabrata and Pichia kudriavzevii, respectively.

We thank you for this information and correction. However, please note that this is a retrospective study conducted through the Hospital electronic record system, thus the terminology refers to previous years.

What techniques were used to identify the species. It is necessary to describe the methods.

The microbiological methods have been added

In the discussion, I think it is important to further explore the expression of cytokines related to septic shock.

Pro-inflammatory cytokines have been further discussed as implicated in the PCT expression pathways and 2 new relevant References have been added (Reference No. 36 and 39). Please note that the corresponding paragraph as well as the References are focusing to the role of cytokines to the PCT levels in the presence of candidemia. (page 11, last paragraph)  

Reviewer 2 Report

Comments and Suggestions for Authors

The findings of this retrospective research are pretty limited.

While the idea is a good one, considering the difficulties in determining cut-off points for different paraclinical findings, especially in ICU patients, and the large amount of literature that analyses the markers evaluated in this paper, my suggestion is to look for more patients that meet the inclusion/exclusion criteria and analyse those results.

Introduction

It covers the topic of the manuscript well. Maybe more emphasis on the lack of evidence for individual markers as discrimination tools between candidemia and bacteriemia might be suitable.

Material and method

The authors should add the time frame of the study.

What is the reasoning for choosing as cut-off point the 25th percentile for the tests and 75th percentile for predicted probabilities? Wouldn't the use of the values of the test be more appropriate, being able in that way to define the best criterion?

Results

If you are reporting specificity, report also the sensibility.

I really suggest building the analysis based on continuous variables, with AOC calculated for each one and for the residuals of the logistic regressions. This will offer a better image of the results (also with charts for those).

Considering the given numbers, the tests have a very low sensitivity. 

Discussions

I suggest to discuss more on potential causes of confusion, considering the results and limitations of this study and the previously published studies.

Line 245-246 you declare that the markers had a high degree of accuracy. Considering the low values for sensitivity (and lack of statistical significance for the difference of the ROC curve from an standard one, this sounds like an overestimation. 

Figure 2 - please redo these to better illustrate the distribution of the values.

Comments on the Quality of English Language

The few present inaccuracies can be solved with a thorough reading.

Author Response

Dear Reviewer,

We would like to thank you for giving us the opportunity to improve our manuscript with your insightful and critical comments. Below, we answer your comments (in bold text) point-by-point.

Introduction

-It covers the topic of the manuscript well. Maybe more emphasis on the lack of evidence for individual markers as discrimination tools between candidemia and bacteriemia might be suitable.

Our answer:

A new sentence has been added to emphasize the limited information regarding the individual markers of infection in discrimination between candidemia and bacteremia (Introduction, 5th paragraph; the new sentence has been highlightened)

Material and method

-The authors should add the time frame of the study.

Our answer:

It has been added (Methods, second line, highlightened)

-What is the reasoning for choosing as cut-off point the 25th percentile for the tests and 75th percentile for predicted probabilities? Wouldn't the use of the values of the test be more appropriate, being able in that way to define the best criterion?

Our answer:

Thank you very much for this crucial comment. We admit that categorizing the tests’ values is statistically inferior to using the continuous values. Therefore, we have performed the statistical analysis from scratch. Specifically, ROC curves were constructed for PCT, CRP, NLR (which were inserted as continuous variables), and the multivariable model, Figure 3. Cut-off values for NLR, PCT, CRP, and the model’s predicted probabilities were estimated by Youden’s index. Then, we calculated all diagnostic characteristics for candidemia prediction for each variable. All these changes are highlighted in the relevant sections of the manuscript, such as in the Statistical analysis (paragraph 2.3) and the Results (paragraphs 3.2 and 3.3) sections. Also, tables 3 and 4 have been totally revised, and a new figure has been added (Figure 3).

Results

-If you are reporting specificity, report also the sensibility.

Our answer:

We assume that you mean sensitivity by the term ‘sensibility’. Every time we report specificity, we also report sensitivity (highlighted in: Results-paragraph 3.3, and Discussion-paragraphs 5,6, and 7).

-I really suggest building the analysis based on continuous variables, with AOC calculated for each one and for the residuals of the logistic regressions. This will offer a better image of the results (also with charts for those).

Our answer:

We appreciate this very insightful comment very much. ROC curves were constructed for PCT, CRP, NLR (which were inserted as continuous variables), and the multivariable model (Figure 3). The model consisted of the continuous values of the three variables. Then, we calculated ROC-AUCs for candidemia prediction for each variable and the model (by using the predicted probabilities of the logistic regression). All these changes are highlighted in the relevant sections of the manuscript, such as in the Results section (paragraphs 3.2 and 3.3). Also, tables 3 and 4 have been totally revised, and a new figure depicting all the AUCs has been added (Figure 3).

-Considering the given numbers, the tests have a very low sensitivity. 

Our answer:

Sensitivities are much higher with the revised statistical analysis, according to your suggestions, for all tests (table 4).

Discussions

-I suggest to discuss more on potential causes of confusion, considering the results and limitations of this study and the previously published studies.

Our answer:

Α sentence has been added in the end of the Limitations paragraph

Line 245-246 you declare that the markers had a high degree of accuracy. Considering the low values for sensitivity (and lack of statistical significance for the difference of the ROC curve from an standard one, this sounds like an overestimation. 

Our answer:

With the revised statistical analysis, according to your suggestions, ROC-AUCs for all tests were statistically significant (Figure 3, table 4), and sensitivities were much higher for all tests (table 4). Anyhow, we do not refer to each marker separately, but to their combination, modeled by the multivariable logistic regression analysis (AUC=0.88, P<0.001), Table 4, Figure 3. However, acknowledging that this is an overestimation, we have rephrased this sentence to: ‘However, this study clearly demonstrates that the combination of easily accessible, everyday clinical laboratory markers had good diagnostic performance for Candida species BSI, which could be important for resource limited regions.’ (page 13 line 1, highlightened)

-Figure 2 - please redo these to better illustrate the distribution of the values.

Our answer:

Thank you for this comment. We have revised both panels (A and B) of Figure 2, by using log transformed values of PCT concentrations.   

Round 2

Reviewer 2 Report

Comments and Suggestions for Authors

I appreciate the changes that were implemented. I consider that the scientific soundness improved significantly.

A few comments to address before the final version:

 - Figure 1 - if possible to be larger because it is impossible to read and it is not clear at all what the AUC represent. Also add legend and notes for Figure 1. The legend of the figure is in contrast with the graphical representation. This is not a boxplot (line 326).

- Figure 2 - it is not necessary 

- Line 194 - text goes around the table

- line 209 - there appear to be two sets of values (some marked with yellow, some without).

- line 2015-2016. Even though the AUC are lower, those still are statistically significant, considering the reported p values.

- Figure 3 seems to be the same with figure 1. Check your results. Add legend for the figure and make it larger.

- PLease be consistent when reporting results. For some you present sensitivity and NPV for other you presents specificity, PPV and LR+. 

- Table 4 should go in the results section

Comments on the Quality of English Language

No significant issues detected.

Author Response

We would like to thank you again for your time to provide us your comments and the important recommendations.

I appreciate the changes that were implemented. I consider that the scientific soundness improved significantly.

A few comments to address before the final version:

 - Figure 1 - if possible to be larger because it is impossible to read and it is not clear at all what the AUC represent. Also add legend and notes for Figure 1. The legend of the figure is in contrast with the graphical representation. This is not a boxplot (line 326).

Answer

We would like to thank you for this important point. In fact, Figure 1 depicts a box plot and corresponds to the legend for Figure 1 (line 278 in the revised submission). Please, find the original Figure 1 below this text. Accidentally, in the Article, the right Figure 1 had been replaced by the Figure 3 (currently, Figure 2 in the revised text, after the removal of the original Figure 2, according to your recommendation). As a result, Figure 3 was shown twice. We apologize for this mistake, probably  made either during the process of our submission or later. 

In the revised manuscript, this figure has replaced the original Figure 1.

- Figure 2 - it is not necessary 

Answer:

This Figure has been removed. Also, the relevant to Figure 2 comment in the Results section, paragraph 3.1 (lines 154-158) ‘The distributions of log transformed PCT concentrations levels in patients with bacteremia and candidemia are shown in Figure 2. Evidently, in patients with candidemia, PCT values were substantially lower compared to mostly within normal range, contrary to patients with bacteremia., in whom PCT values were substantially higher.’ has been removed, too. 

- Line 194 - text goes around the table

Answer

It has been corrected.

- line 209 - there appear to be two sets of values (some marked with yellow, some without).

Answer

Due to mistakes made during the processing of the manuscript to the format used by the journal, in several parts of it there were two sets of values, the new ones marked yellow and the old ones without any marking. The old ones have all been removed throughout the manuscript.

- line 2015-2016. Even though the AUC are lower, those still are statistically significant, considering the reported p values.

Answer

The small P (<0.05) simply means that the Area under the ROC curve (AUC-ROC) is significantly different from 0.5 and, therefore, there is evidence that the laboratory test does have an ability to distinguish between the two groups. However, we wanted to emphasize that the AUC of the model (namely the combination of all 3 variables) was higher than that of each one separately and, as a consequence, it had better diagnostic ability for candidemia compared to each one of them, individually.      

- Figure 3 seems to be the same with figure 1. Check your results. Add legend for the figure and make it larger.

Answer

Thanks to your careful reviewing, it has been corrected (see also our answer to your first comment). In the revised text, Figure 1 shows the Box plot;  Figure 3 has been renamed as Figure 2 after the deletion of Figure 2. It has been made larger, too.

- PLease be consistent when reporting results. For some you present sensitivity and NPV for other you presents specificity, PPV and LR+. 

Answer

Thank you for the comment. We report the full set of diagnostic characteristics, whenever we report such type of results, throughout the manuscript: Results section, paragraph 3.2, highlighted in green. Discussion section, lines 204, 232, 233, 239, 254, highlighted in green.

- Table 4 should go in the results section

Answer

It has been moved to the Results section.

Figure 1

(The figure can not shown here, Please see the uploaded file of our answers to the Reviewer)